# Giving Feedback on Interactive Student Programs with Meta-Exploration

**Evan Zheran Liu**[*], **Moritz Stephan**[*], **Allen Nie, Chris Piech, Emma Brunskill, Chelsea Finn**
Stanford University

## Abstract

Developing interactive software, such as websites or games, is a particularly engaging way to learn computer science. However, teaching and giving feedback on such software is time-consuming — standard approaches require instructors to manually grade student-implemented interactive programs. As a result, online platforms that serve millions, like Code.org, are unable to provide any feedback on assignments for implementing interactive programs, which critically hinders students' ability to learn. One approach toward automatic grading is to learn an agent that interacts with a student's program and explores states indicative of errors via reinforcement learning. However, existing work on this approach only provides binary feedback of whether a program is correct or not, while students require finer-grained feedback on the specific errors in their programs to understand their mistakes. In this work, we show that exploring to discover errors can be cast as a meta-exploration problem. This enables us to construct a principled objective for discovering errors and an algorithm for optimizing this objective, which provides fine-grained feedback. We evaluate our approach on a set of over 700K real anonymized student programs from a Code.org interactive assignment. Our approach provides feedback with 94.3% accuracy, improving over existing approaches by 17.7% and coming within 1.5% of human-level accuracy. Project web page: `https://ezliu.github.io/dreamgrader`.

## 1 Introduction

Feedback plays a critical role in high-quality education, but can require significant time and expertise to provide [7]. We focus on one area where providing feedback is particularly burdensome: modern computer science education, where students are often tasked with developing interactive programs, such as websites or games (e.g., see Figure 1). While developing such programs is highly engaging [35] and has become ubiquitous in contemporary classrooms [12], these programs can include stochastic or creative elements, so they cannot be graded with traditional unit tests and must instead be manually graded. However, such manual grading is increasingly

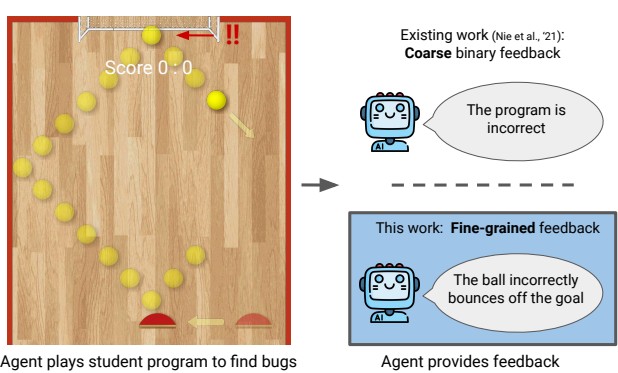

Figure 1: A learned Play-to-Grade agent for the Bounce programming assignment. The agent tests what happens when the ball is hit into the goal, and finds that the ball incorrectly bounces out instead of scoring a point. Whereas prior work provides coarse feedback of whether the program is correct or not, our goal is to provide fine-grained feedback on the specific mistakes a student has made.

---

[*]Co-first authors. Correspondence to `evanliu@cs.stanford.edu`.

36th Conference on Neural Information Processing Systems (NeurIPS 2022).

infeasible with the growing demand for computer science education and rise of massive online learning platforms. For example, one popular platform, Code.org has enrolled over 70M students [7]. As manually grading a submission can take up to 6 minutes, a single assignment creates decades of grading labor. Consequently, platforms like Code.org cannot yet provide feedback about whether an interactive assignment submission is correct or not, let alone more fine-grained feedback.

To alleviate this enormous grading burden, Nie et al. [30] introduce the Play-to-Grade paradigm for automatically providing feedback by training a reinforcement learning agent to grade a program the same way humans do: by interacting or playing with the program. The idea is for the agent to visit states that reveal errors in the program, and then aggregate this information as feedback. Such an agent is trained on a set of training programs labeled with feedback (e.g., provided by an instructor), and the goal is to generalize to new student programs. Figure 1 shows an example learned agent that tests what happens when the ball is hit into the goal, exposing an error where the ball bounces off the goal instead of entering and scoring a point. The state-of-the-art approach in this paradigm provides accurate coarse feedback of whether the program is completely correct or not [30]. However, to understand their mistakes, students usually require more specific feedback about what errors are in their programs.

Learning an agent to explore and discover the errors in a program to provide such fine-grained feedback is challenging: Most errors cannot be discovered with simple random exploration, and instead require targeted exploration, such as deliberately hitting the ball into the goal. In addition, the agent must be able to adapt its exploration to different programs, which each behave differently and may present unexpected obstacles, such as multiple balls. Our key technical insight is that learning to discover errors connects with the meta-exploration problem in meta-reinforcement learning (meta-RL). This insight enables us to leverage techniques from the meta-exploration literature to construct and optimize a principled objective for producing fine-grained feedback. Specifically, we follow the Play-to-Grade paradigm and assume access to 3,556 training programs, labeled with the errors in the program. Then, we formulate the problem as maximizing the mutual information between states visited by our agent and the label. Finally, we use techniques from the DREAM meta-RL algorithm [25] to decompose this objective into shaped rewards that enable learning sophisticated exploration.

Overall, the main contribution of this work is to connect the Play-to-Grade paradigm with the meta-RL literature, and consequently, to provide an effective method for providing fine-grained feedback for interactive student programs, which we call DREAMGRADER. Additionally, we release our code to propose automatic feedback as a new meta-RL testbed that fulfills an unmet need in the community for a benchmark that is simultaneously readily accessible and directly impactful. We evaluate our system on 711,274 anonymized student submissions of the Bounce assignment from Code.org [30]. Trained on 3,556 programs, DREAMGRADER achieves an accuracy of 94.3%, which improves over existing approaches by 17.7% and comes within 1.5% of human-level grading accuracy. In addition, our approach can significantly reduce instructor grading burden: while manually grading all student submissions would require 4 years of work, our system can grade the same set of programs $180\times$ faster on a single GPU and can be parallelized for even faster grading over multiple GPUs.

## 2    Related Works

**Educational feedback.** We consider the problem of automatically providing feedback, which plays an important role in student learning and motivation [32]. Though we specifically focus on feedback, other works also leverage machine learning for other aspects of education, including tracking what students know [46, 8, 36], predicting student retention [9, 4, 41, 3], and building intelligent tutoring systems [1, 31]. Work on automatically providing feedback for computer science assignments focuses on two main approaches: analyzing either (i) the code or (ii) the behavior of a program. Methods that analyze code provide feedback by passing the code through a neural network [37, 6, 26, 51], or by constructing syntax trees [43, 49], e.g., to predict useful next implementation steps [39, 33]. Analyzing code works well for shorter programs (e.g., under 50 lines of code) and has even been deployed in online courses [52]. However, this approach struggles to scale to lengthier or more complex programs. Hence, we instead opt for the second approach of analyzing program behavior, which conveniently does not depend on program length, though it requires the program to be executable.

Arguably, the simplest method of analyzing program behavior is unit testing. Unit testing can provide automatic feedback to some extent when the desired output of each input is known, but this is typically not the case with interactive programs, such as websites or games. Instead, work on

automated testing can provide feedback by generating corner-case inputs that reveal errors via input fuzzing [14], symbolic generation [21], or reinforcement learning exploration objectives [54, 15]. However, this line of work assumes that errors are easy to detect when revealed, while detecting a revealed error itself can be challenging [30].

Consequently, Nie et al. [30] propose the Play-to-Grade paradigm to both learn an agent to discover states that reveal errors, and a model to detect and output the revealed errors. Our work builds upon the Play-to-Grade paradigm, but differs from Nie et al. [30] in the provided feedback. While Nie et al. [30] only provide coarse binary feedback of whether a program is correct, we introduce a new principled objective to provide fine-grained feedback of what specific errors are present to help students understand their mistakes.

**Meta-reinforcement learning.** To provide fine-grained feedback, we connect the problem of discovering errors with the meta-exploration problem in meta-RL. There is a rich literature of approaches that learn to explore via meta-RL [16, 44, 38, 40, 55, 56, 19, 17, 20, 25]. We specifically leverage ideas from the DREAM algorithm [25] to construct a shaped reward function for learning exploration. Our work has two key differences from prior meta-RL research. First, we introduce a novel factorization of the DREAM objective that better generalizes to new programs. Second, and more importantly, we focus on the problem of providing feedback on interactive programs. This differs from a large body of meta-RL work that focuses on interesting, yet synthetic problems, such as 2D and 3D maze navigation [10, 27, 56, 25], simulated control problems [11, 38, 53], and alchemy [48]. While meta-RL has been applied to realistic settings in robotics [29, 2, 42], such application requires costly equipment. In contrast, this work provides a new meta-RL problem that is both realistic and readily accessible.

## 3 The Fine-Grained Feedback Problem

We consider the problem of automatically providing feedback on programs. During training, we assume access to a set of programs labeled with the errors made in the program (i.e., ground-truth instructor feedback). During testing, the grading system is presented with a new student program and must output feedback of what errors are in the program. To produce this feedback, the grading system is allowed to interact with the program.

More formally, we consider a distribution over programs $p(\mu)$, where each program $\mu$ defines a Markov decision process (MDP) $\mu = \langle \mathcal{S}, \mathcal{A}, \mathcal{T}, \mathcal{R} \rangle$ with states $\mathcal{S}$, actions $\mathcal{A}$, dynamics $\mathcal{T}$, and rewards $\mathcal{R}$. We assume that the instructor creates a *rubric*: an ordered list of $K$ potential errors that can occur in a program. Each program $\mu$ is associated with a ground-truth label $y \in \{0, 1\}^K$ of which errors are made in the program. The $k^{\text{th}}$ index $y_k$ denotes that the $k^{\text{th}}$ error of the rubric is present in the program $\mu$.

During training, the grading system is given a set of $N$ labeled training programs $\{(\mu^n, y^n)\}_{n=1}^N$. The goal is to learn a *feedback function* $f$ that takes a program $\mu$ and predicts the label $\hat{y} = f(\mu)$ to maximize the *expected grading accuracy* $\mathcal{J}_{\text{grade}}(f)$ over test programs $\mu$ with *unobserved* labels $y$:

$$\mathcal{J}_{\text{grade}}(f) = \mathbb{E}_{\mu \sim p(\mu)} \left[ \frac{1}{K} \sum_{k=1}^K \mathbb{I}[f(\mu)_k = y_k] \right], \tag{1}$$

where $\mathbb{I}$ is an indicator variable. Effectively, $\mathcal{J}_{\text{grade}}$ measures the per-rubric item accuracy of predicting the ground-truth label $y$. To predict the label $y$, the feedback function may optionally interact with the MDP $\mu$ defined by the program for any small number of episodes.

**Bounce programming assignment.** Though the methods we propose in this work generally apply to any interactive programs with instructor-created rubrics and we include experiments on another interactive assignment in Appendix D.2, we primarily focus on the Bounce programming assignment from Code.org, a real online assignment that has been completed nearly a million times. As providing feedback for interactive assignments is challenging, the assignment currently provides no feedback whatsoever on Code.org, and instead relies on the student to discover their own mistakes by playing their program. This assignment is illustrated in Figure 1. Each student program defines an MDP, where we use the state representation from Nie et al. [30]: each state consists of the $(x, y)$-coordinates of the paddle and balls, as well as the $(x, y)$-velocities of the balls. There are three actions: moving the paddle left or right, or keeping the paddle in the current position. In the dynamics of a correct program,

the ball bounces off the paddle and wall. When the ball hits the goal or floor, it disappears and launches a new ball, which increments the player score and opponent score respectively. However, the student code may define other erroneous dynamics, such as the ball passing through the paddle or bouncing off the goal. The reward is $+1$ when the player score increments and $-1$ when the opponent score increments. An episode terminates after $100$ steps or if either the player or opponent score exceeds $30$.

Our experiments use a dataset of 711,274 real anonymized student submissions to this assignment, released by Nie et al. [30]. We use $0.5\%$ of these programs for training, corresponding to $N = 3{,}556$ and uniformly sample from the remaining programs for testing.

Possible errors in a student program take the form of "when *event* occurs, there is an incorrect *consequence*," where the list of all events and consequences is listed in Table 1. For example, Figure 1 illustrates the error where the event is the ball hitting the goal, and the consequence is that the

Table 1: Possible event and consequence types of program errors.

| Event | | Consequence |
|-------|---|-------------|
| Ball hits paddle | | Ball bounces / does not bounce |
| Ball hits wall | | Increments / does not increment player score |
| Ball hits goal | $\times$ | Increments / does not increment opponent score |
| Ball hits floor | | Launches / does not launch a new ball |
| Paddle moves | | Moves the paddle |
| Program starts | | |

ball incorrectly bounces off the goal, rather than entering the goal. For simplicity, we primarily consider a representative rubric of $K = 8$ errors, spanning all event and consequence types, listed in Appendix A, though we include some experiments on all error types in Appendix D.

**Prior approach for program feedback.** Accurately determining which errors are in a given program is challenging, because it requires targeted exploration that adapts to variability in the programs. Often, the presence of some errors makes it difficult to find other errors, such as multiple balls making it difficult to determine which events change the score. Prior work by Nie et al. [30] sidesteps this challenge by only providing coarse feedback of whether a program is correct or not, by determining if a student program differs from a reference solution program. Such coarse feedback is much easier to provide, as it often involves only finding the most obvious error, which can frequently be found with relatively untargeted exploration. In the next section, we present a new approach that instead targets exploration toward uncovering specific misconceptions to effectively provide fine-grained feedback. We discuss how our approach differs from Nie et al. [30] in greater detail in Appendix E.

## 4 Automatically Providing Fine-Grained Feedback with DREAMGRADER

In this section, we detail our approach, DREAMGRADER, for automatically providing fine-grained feedback to help students understand their mistakes. From a high level, DREAMGRADER learns two components that together form the feedback function $f$:

(i) An *exploration policy* $\pi$ that acts on a program $\mu$ to produce a trajectory $\tau = (s_0, a_0, r_0, \ldots)$.

(ii) A *feedback classifier* $g(y \mid \tau)$ that defines a distribution over labels $y$ given a trajectory $\tau$.

The idea is to explore states that either indicate or rule out errors with the exploration policy, and then summarize the discovered errors with the feedback classifier. To provide feedback on a new program $\mu$, we first roll out the exploration policy $\pi$ on the program to obtain a trajectory $\tau$, and then obtain the predicted label $\arg \max_y g(y \mid \tau)$ by applying the feedback classifier. Under this parametrization of the feedback function $f$, we can rewrite the expected grading accuracy objective in Equation 1 as:

$$\mathcal{J}_{\text{DREAMGRADER}}(\pi, g) = \mathbb{E}_{\mu \sim p(\mu), \tau \sim \pi(\mu)} \left[ \frac{1}{K} \sum_{k=1}^{K} \mathbb{I}[\arg \max_{\hat{y}} g(\hat{y} \mid \tau)_k = y_k] \right], \qquad (2)$$

where $\pi(\mu)$ denotes the distribution over trajectories from rolling out the policy $\pi$ on the program $\mu$.

After this rewriting of the objective, our approach is conceptually straightforward: we learn both the exploration policy and classifier to maximize our rewritten objective. We can easily learn the feedback classifier $g$ by maximizing the probability of the correct label given a trajectory generated by the exploration policy with standard supervised learning (i.e., cross-entropy loss), but learning the exploration policy $\pi$ is more challenging. Note that we could directly optimize our objective in Equation 2 by treating the inside of the expectation as a reward received at the end of the episode and use this to learn the exploration policy $\pi$ with reinforcement learning. However, this reward

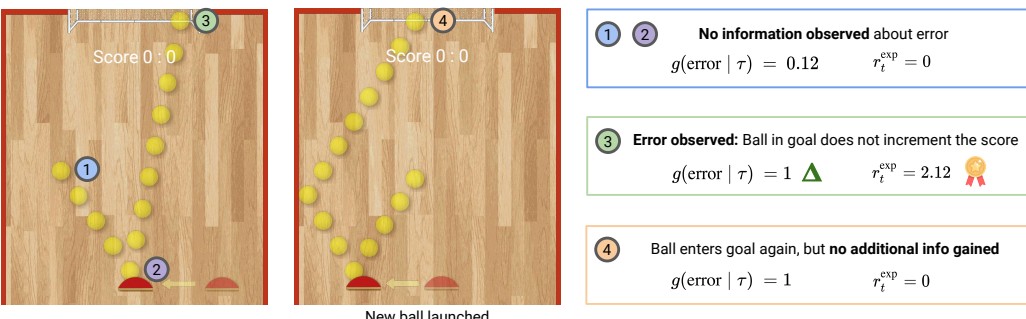

Figure 2: DREAMGRADER provides credit assignment for learning exploration by leveraging the feedback classifier $g(y \mid \tau)$. Here, we consider the error "when the ball enters the goal, the player score does not increment." At ① and ②, no information is observed that either rules out or indicates the error. Hence, no exploration reward $r_t^{\text{exp}}$ is provided, and the classifier assigns $0.12$ probability that the error is present, reflecting the prior that 12% of the training programs have this error. At ③, the ball enters the goal, but does not score a point and a new ball is launched. This indicates that the error is present, so the classifier updates, which creates high exploration reward and credit assignment for learning exploration. At ④, the ball enters the goal again. However, no additional information is gained, so the classifier does not change, and no reward is given. Overall, this enables learning effective exploration that purposely hits the ball into the goal once.

signal makes credit assignment difficult for learning the exploration policy, since it is given at the end of the episode, rather than at the states where the exploration policy discovers errors. Indeed, we empirically find that learning from this reward signal struggles to adequately explore (Section 5).

Hence, our goal is instead to construct a reward signal that helps assign credit for the exploration policy and provides rewards at the states that indicate or rule out errors in the program. To do this, we first propose an alternative objective that is sufficient to maximize the Play-to-Grade objective, but can be decomposed into per-timestep rewards that correctly assign credit (Section 4.1). Intuitively, these rewards leverage the feedback classifier to provide high reward when an action leads to a state that increases the classifier's certainty about whether an error is present. Then, we detail practical design choices for implementing this approach with neural networks (Section 4.2) and conclude by drawing a connection between learning to find errors and the meta-exploration problem, which motivates our choice of alternative objective and its subsequent decomposition (Section 4.3).

## 4.1 Assigning Credit to Learn Exploration

We now obtain a reward signal to help assign credit for learning exploration. From a high level, we first propose to maximize a mutual information objective that is sufficient to maximize our objective $\mathcal{J}_{\text{DREAMGRADER}}(\pi, g)$ in Equation 2. We then rewrite our mutual information objective in terms of per-timestep exploration rewards related to the information gain of the feedback classifier on the true label $y$ when it observes the transition $(s_t, a_t, r_t)$. This helps assign credit for learning the exploration policy, as the transitions that either indicate or rule out errors in the program are exactly those that have high information gain. Figure 2 illustrates an example of the derived exploration rewards.

**Objective.** Intuitively, we want our exploration policy to visit states that either indicate or rule out errors. We can formalize this intuition by maximizing the mutual information $I(\tau; y)$ between the trajectories $\tau \sim \pi$ visited by the policy and the feedback label of the program $y$, which causes the policy to visit states that make the feedback label highly predictable. Importantly, maximizing this objective is sufficient to maximize the expected grading accuracy $\mathcal{J}_{\text{DREAMGRADER}}(\pi, g)$ in Equation 2: Let $p(y \mid \tau)$ be the true posterior over labels given trajectories $\tau \sim \pi$ sampled from the policy. Maximizing the mutual information $I(\tau; y)$ produces trajectories that maximize the probability of the label under the true posterior $p(y \mid \tau)$. Then, if the feedback classifier is learned to match the true posterior $g(y \mid \tau) = p(y \mid \tau)$ while $I(\tau; y)$ is maximized, the expected grading accuracy $\mathcal{J}_{\text{DREAMGRADER}}(\pi, g)$ is also maximized.

**Optimization.** We can efficiently maximize our objective $I(\tau; y)$ by maximizing a variational lower bound [5] and decomposing the lower bound into a per-timestep reward that helps assign credit for learning the exploration policy $\pi$. Below, we derive this for the case of learning only a single exploration policy $\pi$ to uncover all errors, though we will later discuss how we can factorize this to

learn $N$ exploration policies $\{\pi_i\}_{i=1}^N$ that each explore to uncover a single error type.

$$I(\tau; y) = H[y] - H[y \mid \tau] \tag{3}$$

$$= H[y] + \mathbb{E}_{\mu \sim p(\mu), \tau \sim \pi(\mu)} \left[ \log p(y \mid \tau) \right] \tag{4}$$

$$\geq H[y] + \mathbb{E}_{\mu \sim p(\mu), \tau \sim \pi(\mu)} \left[ \log g(y \mid \tau) \right] \tag{5}$$

$$= H[y] + \mathbb{E}_{\mu \sim p(\mu), \tau \sim \pi(\mu)} \left[ \log g(y \mid s_0) + \sum_{t=0}^{T-1} r_t^{\text{exp}} \right], \tag{6}$$

where $r_t^{\text{exp}} = \log g(y \mid \tau_{:t+1}) - \log g(y \mid \tau_{:t})$ and
$T$ is the length of the trajectory $\tau = (s_0, a_0, r_0, \ldots, s_T)$.

The inequality in (5) holds for replacing the true posterior $p(y \mid \tau)$ with any distribution, and (6) comes from expanding a telescoping series as done by DREAM [25], where $\tau_{:t} = (s_0, a_0, r_0, \ldots, s_t)$ denotes the the trajectory up to the $t^{\text{th}}$ state.

This derivation provides a shaped reward function $r_t^{\text{exp}}$ for learning the exploration policy. Intuitively, this reward captures how much new information the transition $(s_t, a_t, r_t, s_{t+1})$ provides to the feedback classifier $g$ on what errors are in the program: The reward is high if observing this transition either indicates an error (e.g., the ball enters the goal, but does not score a point) or rules out an error (e.g., the ball enters the goal and scores a point), and is low otherwise.

Additionally, we now have a recipe for maximizing the mutual information $I(\tau; y)$ to learn both the policy $\pi$ and feedback classifier $g$. Only the second term in (6) depends on $\pi$ and $g$. Hence, we can maximize this lower bound on $I(\tau; y)$ by maximizing the log-likelihood of the label with respect to the feedback classifier $\mathcal{J}_{\text{feedback}}(g) = \mathbb{E}_{\mu \sim p(\mu), \tau \sim \pi(\mu)} \left[ \log g(y \mid \tau) \right]$, and maximizing the rewards $r_t^{\text{exp}} = \log g(y \mid \tau_{:t+1}) - \log g(y \mid \tau_{:t})$ with respect to the policy $\pi$ via reinforcement learning.

**Factorizing our objective.** So far, our approach learns a single exploration policy that must uncover all error types in the rubric. However, learning such a policy can be challenging, especially if uncovering different error types requires visiting very different states. We instead propose to learn a separate exploration policy $\pi_k$ for each error index of the rubric $k = 1, \ldots, K$. We can accomplish this by observing that maximizing the mutual information $I((\tau_1, \ldots, \tau_K); y)$ between $K$ trajectories $\tau_i \sim \pi_i$ is also sufficient to maximize the expected grading accuracy. Furthermore, maximizing the mutual information with each dimension of the label $I(\tau_k; y_k)$, where $\tau_k \sim \pi_k$ for each $k$, is sufficient to maximize the mutual information with the entire label $I((\tau_1, \ldots, \tau_K); y)$. We therefore can derive exploration rewards for each term $I(\tau_k; y_k)$ to learn each policy $\pi_k$ with rewards $r_t^{\text{exp}} = \log g(y_k \mid \tau_{:t+1}) - \log g(y_k \mid \tau_{:t})$. We find that this improves grading accuracy in our experiments (Section 5) and enables parallel training and testing for the $K$ exploration policies.

### 4.2 A Practical Implementation

Overall, DREAMGRADER consists of a feedback classifier $g$ and $K$ exploration policies $\{\pi_k\}_{k=1}^K$, where the $k^{\text{th}}$ policy $\pi_k$ tries to visit states indicative of whether the $k^{\text{th}}$ error type is present in the program. We learn these components by repeatedly running training episodes for each policy $\pi_k$ with Algorithm 1. We first sample a labeled training program and follow the policy on the program (lines 1–

---

**Algorithm 1** Training episode for policy $\pi_k$

1: Sample a training program $\mu$ with label $y$
2: Roll out policy to obtain trajectory $\tau \sim \pi_k(\mu)$
3: Compute rewards with feedback classifier
   $r_t^{\text{exp}} = \log g(y_k \mid \tau_{:t+1}) - \log g(y_k \mid \tau_{:t})$
4: Update policy to maximize rewards $r_t^{\text{exp}}$ with RL
5: Update feedback classifier to max. $\log g(y_k \mid \tau)$

---

2). Then, we maximize our mutual information objective by updating the policy with our exploration rewards (lines 3–4), and by updating the classifier to maximize the log-likelihood of the label (line 5).

In practice, we parametrize the exploration policies and feedback classifier as neural networks. Since the exploration rewards $r_t^{\text{exp}}$ depend on the past and are non-Markov, we make each exploration policy $\pi_k$ recurrent: At timestep $t$, the policy $\pi_k(a_t \mid (s_0, a_0, r_0, \ldots, s_t))$ conditions on all past states, actions, and observed rewards for each $k$. We parametrize each policy as a deep dueling double Q-networks [28, 50, 45]. Consequently, our policy updates in line 4 consist of placing the trajectory in a replay buffer with rewards $r_t^{\text{exp}}$ and sampling from the replay buffer to perform Q-learning updates. We parametrize each dimension of the feedback classifier $g(y_k \mid \tau)$ for $k = 1, \ldots, K$ as a separate neural network. We choose not to share parameters between the exploration policies and between

the dimensions of the feedback classifier for simplicity. However, we note that significant parameter sharing is likely possible, as exploration policies for two different error types can be extremely similar (e.g., two errors that involve hitting the ball into the goal). Automatically determining which parameters to share could be an interesting direction for future work, as it is not known a priori which error types are related to each other. See Appendix B for full architecture and model details.

### 4.3 Play-to-Grade as Meta-Exploration

Our choice to optimize the mutual information objective $I(\tau; y)$ and decompose this objective into per-timestep rewards using techniques from the DREAM meta-RL algorithm stems from the fact that the Play-to-Grade paradigm can be cast as a meta-exploration problem. Specifically, meta-RL aims to learn agents that can quickly learn new tasks by leveraging prior experience on related tasks. The standard few-shot meta-RL setting formalizes this by allowing the agent to train on several MDPs (tasks). At time time, the agent is presented with a new MDP and is allowed to first explore the new MDP for several episodes (i.e., the few shots) to gather information. Then, it must use the information it gathered to solve the MDP and maximize returns on new episodes. Learning to efficiently spend these allowed few exploration episodes to best solve the test MDP is the meta-exploration problem.

Our setting of providing feedback follows the exact same structure as few-shot meta-RL. In our setting, we can view each student program as a new 1-step task of predicting the feedback label, where the reward is the number of dimensions of the label that are correctly predicted. To predict this label, the Play-to-Grade paradigm first explores the program for several episodes to discover states indicative of errors, which corresponds to the few shots. The key challenge in this setting is exactly how to best spend those few exploration episodes: i.e., to gather the information needed to predict the label, which is exactly the meta-exploration problem. This bridge between identifying errors in programs and meta-exploration suggests that techniques from each body of literature could mutually benefit each other. Indeed, DREAMGRADER leverages ideas from DREAM and future work could explore other techniques to transfer across the two areas. Additionally, this connection also offers a new testbed for meta-exploration and meta-RL research. As discussed in Section 2, while existing meta-RL benchmarks tend to be either readily accessible or impactful and realistic, automatically providing feedback simultaneously provides both, and we release code for a meta-RL wrapper of the Bounce programming assignment to spur further research in this direction.

## 5 Experiments

In our experiments, we aim to answer five main questions: (1) How does automated feedback grading accuracy compare to human grading accuracy? (2) How does DREAMGRADER compare with Nie et al. [30], the state-of-the art Play-to-Grade approach? (3) What are the effects of our proposed factorization and derived exploration rewards on DREAMGRADER? (4) How much human labor is saved by automating feedback? (5) Interactive programs can be particularly challenging to grade because test programs can contain behaviors not seen during training — how well do automated feedback systems generalize to such unseen behaviors? To answer these questions we consider the dataset of 700K real anonymized Bounce student programs, described in Section 3.

Below, we first establish the points of comparison to necessary answer these questions (Section 5.1). Then, we evaluate these approaches to answer the first four questions (Section 5.2). Finally, we answer question (5) by evaluating DREAMGRADER on variants of Bounce student programs that modify the ball and paddle speeds, including speeds not seen during training (Section 5.3). Additionally, in Appendix D, we test if DREAMGRADER can scale to all error types and an additional interactive assignment called Breakout, which is widely taught in university and highschool classrooms.

### 5.1 Points of Comparison

We compare with the following four approaches. Unless otherwise noted, we train 3 seeds of each automated approach for 5M steps on $N = 3{,}556$ training programs, consisting of $0.5\%$ of the dataset.

**Human grading.** To measure the grading accuracy of humans, we asked for volunteers to grade Bounce programming assignments. We obtained 9 volunteers consisting of computer science undergraduate and PhD students, 7 of whom had previously instructed or been a teaching assistant for a

computer science course. Each volunteer received training on the Bounce programming assignment and then was asked to grade 6 randomly sampled Bounce programs. See Appendix C.1 for details.

**Nie et al. [30] extended to provide fine-grained feedback.** We extend the original Play-to-Grade approach, which provides binary feedback about whether a program is completely correct or not, to provide fine-grained feedback. Specifically, Nie et al. [30] choose a small set of 10 training programs, curated so that each program exhibits a single error, and together, they span all errors. Then, for each training program, the approach learns (i) a distance function $d(s, a)$ that takes a state-action tuple $(s, a)$, trained to be large for tuples from the buggy training program and small for tuples from a correct reference implementation; and (ii) an exploration policy that learns to visit state-actions where $d(s, a)$ is large. To provide feedback to a new student program, the approach runs each exploration policy on the program and outputs that the program has an error if any of the distance functions is high for any of the tuples visited by the exploration policies.

We extend this approach to provide fine-grained feedback by following a similar set up. We follow the original procedure to train a separate policy and distance function on $K = 8$ curated training programs, where the $k^{\text{th}}$ program exhibits only the $k^{\text{th}}$ error from the rubric we consider. Then, to provide fine-grained feedback on a new program, we run each policy on the new program and predict that the $k^{\text{th}}$ error is present (i.e., $\hat{y}_k = 1$) if the $k^{\text{th}}$ distance function is high on any state-action tuple. We use code released by the authors without significant fine-tuning or modification. We emphasize that this approach only uses 8 curated training programs, as opposed to the $N = 3,556$ randomly sampled programs used by other automated approaches, as this approach is not designed to use more training programs, and furthermore cannot feasibly scale to many more training programs, as it learns a distance function and policy for every training program.

**DREAMGRADER (direct max).** To study the effect of our derived exploration rewards $r_t^{\text{exp}}$, we consider the approach of directly maximizing the DREAMGRADER objective in Equation 2, described at the beginning of Section 4.1. This approach treats the inside of the expectation as end-of-episode returns, and does not provide explicit credit assignment. This approach is equivalent to maximizing the DREAMGRADER objective with the $\text{RL}^2$ meta-RL algorithm [10, 47].

**DREAMGRADER (unfactorized).** Finally, to study the effect of our proposed factorization scheme, described at the end of Section 4.1, we consider a variant of DREAMGRADER where we do not factorize the objective and instead only learn a single exploration policy to uncover all errors.

## 5.2 Main Results

We compare the approaches based on grading accuracy, precision, recall, and F1 scores averaged across the 8 error types in the rubric. Table 2 summarizes the results. Overall, we find that DREAMGRADER achieves

Table 2: Accuracy, precision, recall and F1 of grading systems, averaged across the $K = 8$ errors of the rubric, with 1-standard deviation error bars.

|  | Accuracy | Precision | Recall | F1 |
|---|---|---|---|---|
| Human | **95.8 ± 3.9%** | **95.0 ± 13.2%** | **91.1 ± 10.0%** | **91.9 ± 8.3%** |
| DREAMGRADER | 94.3 ± 1.3% | 76.7 ± 5.8% | **94.3 ± 1.6%** | 84.6 ± 1.5% |
| DREAMGRADER (unfactorized) | 91.3 ± 0.4% | 72.9 ± 0.5% | 68.9 ± 1.0% | 70.8 ± 0.7% |
| DREAMGRADER (direct max) | 84.8 ± 2.2% | 36.3 ± 1.7% | 37.8 ± 9.7% | 36.6 ± 5.1% |
| Nie et al. [30] | 75.5 ± 0.9% | 24.9 ± 5.0% | 27.7 ± 7.1% | 26.1 ± 5.7% |

the highest grading accuracy of the automated grading approaches, providing feedback with 17.7% greater accuracy than Nie et al. [30]. Furthermore, DREAMGRADER comes within 1.5% of human-level grading accuracy. DREAMGRADER achieves this by learning exploration behaviors that probe each possible error event (see Appendix D.3 or https://ezliu.github.io/dreamgrader for visualizations of the learned behaviors).

**Analysis.** To further understand these results, we plot the training curves of the grading accuracy on each error type in the rubric vs. the number of training steps of DREAMGRADER, as well as the final grading accuracies of the other approaches in Figure 3. The performance of variants of DREAMGRADER underscores the importance of our design choices: DREAMGRADER (direct max) achieves significantly lower accuracy than DREAMGRADER across all error types, indicating the importance of our shaped exploration rewards $r_t^{\text{exp}}$ for learning effective exploration. Additionally, while DREAMGRADER (unfactorized) achieves relatively high average accuracy, it still performs worse than DREAMGRADER. This illustrates the difficulty of learning a single exploration policy to uncover all errors, which is alleviated by our factorization.

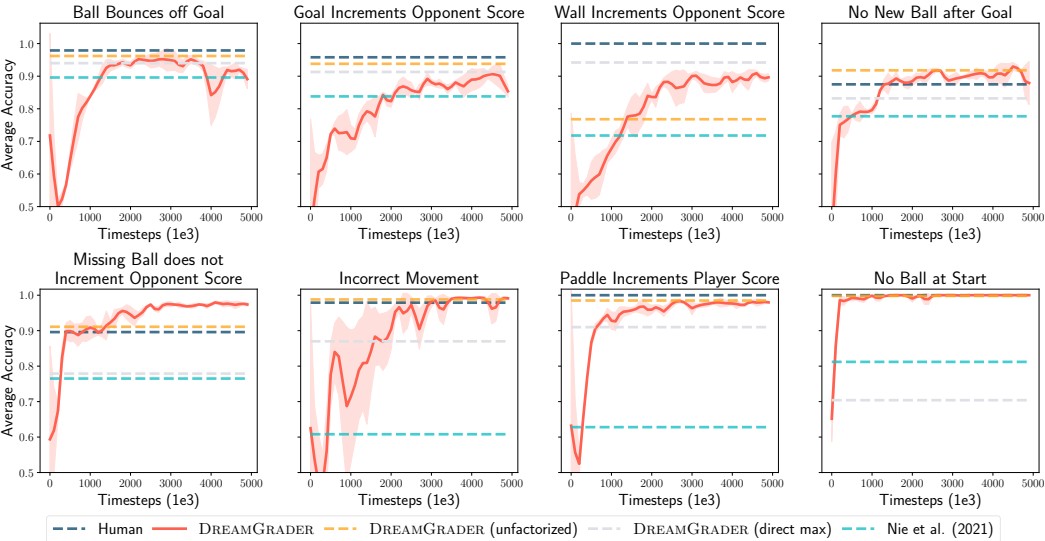

Figure 3: Average grading accuracy for each error type vs. number of training steps for DREAMGRADER with 1-stddev error bars. We plot the final grading accuracies of the other approaches as horizontal lines.

Additionally, we find that DREAMGRADER achieves human-level grading accuracy on 5 of the 8 error types, but relatively struggles on the 3 errors on the left of the top row of Figure 3, which leads to overall lower average grading accuracy. For the errors involving incrementing the opponent score, we qualitatively find that DREAMGRADER most commonly struggles when there are multiple balls, which makes it difficult to ascertain which events are causing the opponent score to increment. Human grading was able to circumvent this issue, but humans required playing the program for up to 40 episodes, while we limited DREAMGRADER to a single episode per error type.

**Reduction in human grading burden.** We found that our grading volunteers required between 1–6 minutes to grade each program, averaging around 3 minutes. At this grading speed, grading all of the 700K Code.org submissions would take close to 4 years of human labor. In contrast, automatic grading with DREAMGRADER requires only 1 second per program on a single NVIDIA RTX 2080 GPU, which is $180\times$ faster.

## 5.3 Generalizing to Unseen Behaviors

A key challenge in providing feedback for interactive programs is that student programs may include behaviors not seen during training. To evaluate generalization to unseen behaviors, we consider varying the speeds of the ball and paddle. There are five possible settings for the ball and paddle speeds: very slow, slow, normal, fast, and very fast. To test the ability of DREAMGRADER to generalize to unseen behaviors at test time, we train DREAMGRADER on programs where we hold out the normal ball and paddle speeds, and then test DREAMGRADER on programs with the held out speeds. Specifically, we use the same $N = 3{,}556$ training programs as before, but we uniformly randomize the ball and paddle speeds independently to be one of the four non-held out speeds. Then we evaluate on test programs with (1) held out ball and paddle speeds; (2) held out ball speed with random training paddle speed; (3) held out paddle speed with random training ball speed; and (4) training ball and paddle speeds.

DREAMGRADER generalizes to unseen ball and paddle speeds at test time. Compared to the standard training in the previous section, where all ball and paddles had the "normal" speed, performance drops, as under faster balls, certain behaviors like hitting the ball into the goal cannot be reliably achieved, and increasing the training data does not decrease the performance drop. However, accuracy remains relatively high, and DREAMGRADER performs about the same on

Table 3: DREAMGRADER's results under held out ball and paddle speeds. DREAMGRADER generalizes to ball and paddle speeds not seen during training.

|  | Both held out | Held out ball speed | Held out paddle speed | No held out speed |
|---|---|---|---|---|
| Accuracy | $89.0 \pm 1.9\%$ | $89.7 \pm 1.7\%$ | $89.6 \pm 2.0\%$ | $89.3 \pm 1.8\%$ |
| Precision | $40.0 \pm 1.0\%$ | $44.1 \pm 1.8\%$ | $46.2 \pm 1.3\%$ | $41.6 \pm 2.1\%$ |
| Recall | $88.2 \pm 2.3\%$ | $89.3 \pm 1.5\%$ | $89.0 \pm 1.7\%$ | $85.5 \pm 2.4\%$ |
| F1 | $55.0 \pm 1.3\%$ | $59.1 \pm 1.9\%$ | $60.1 \pm 0.8\%$ | $56.0 \pm 2.4\%$ |

test programs regardless of whether the ball and paddle speeds were seen during training or not. This indicates some ability to generalize to unseen behaviors at test time. Table 3 displays the full results.

# 6 Conclusion

In this work, we introduced DREAMGRADER, an automatic grading system for interactive programs that provides fine-grained feedback at near human-level accuracy. The key insight behind our system is connecting the problem of automatically discovering errors with the meta-exploration problem in meta-RL, which yields important benefits for both sides. On the one hand, this connection offers a powerful and previously unexplored toolkit to computer science education, and more generally, to discovering errors in websites or other interactive programs. On the other hand, this connection also opens impactful and readily accessible applications for meta-RL research, which has formerly primarily focused on synthetic tasks due to the lack of more compelling accessible applications.

While DREAMGRADER nears human-level grading accuracy, we caution against blindly replacing instructor feedback with automated grading systems, such as DREAMGRADER, which can lead to potentially negative educational and societal impacts without conscientious application. For example, practitioners must ensure that automated feedback is equitable and not biased against certain classes of solutions that may be correlated with socioeconomic status. One option to mitigate the risk of potential negative consequences while still reducing instructor labor is to use automated feedback systems to *assist* instructors similar to prior work [13], e.g., by querying the instructor on examples where the system has low certainty, or presenting videos of exploration behavior from the system for the instructor to provide the final feedback.

Finally, this work takes an important step to reduce teaching burden and improve education, but we also acknowledge DREAMGRADER still has important limitations to overcome. Beyond the remaining small accuracy gap between DREAMGRADER and human grading, DREAMGRADER also requires a substantial amount of training data. While we only use 0.5% of the Bounce dataset for training, it still amounts to 3,556 labeled training programs, and labeling this many programs can be prohibitive for smaller-scale classrooms, though feasible for larger online platforms. We hope that our release of Bounce as a meta-RL problem can help spur future work to overcome these limitations.

**Reproducibility.** Our code is publicly available at `https://github.com/ezliu/dreamgrader`, which includes Bounce as a new meta-RL testbed.

## Acknowledgments and Disclosure of Funding

We thank Kali Stover from the Institutional Review Board (IRB) and Ruth O'Hara and Tallie Wetzel from the Student Data Oversight Committee (SDOC) for reviewing our process of asking humans to grade programs. We thank our human graders: Annie Xie, Sahaana Suri, Cesar Lema, Olivia Lee, Moritz Stephan, Kaien Yang, Maximilian Du, Patricia Strutz, and Govind Chada.

We thank Annie Xie and Sahaamble Suri for thwarting EL's best attempts at procrastination, without whom, this work would not have been completed in time.

EL is supported by a National Science Foundation Graduate Research Fellowship under Grant No. DGE-1656518. CF is a Fellow in the CIFAR Learning in Machines and Brains Program. This work was also supported in part by Google, Intel, and a Stanford Human-Centered AI Hoffman Yee grant. Icons in this work were made by FreePik from FlatIcon.

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
