# A Dataset Details

We use the Bounce programming assignment dataset from Code.org, released by Nie et al. [30]. We release code that packages this dataset as an easy-to-use meta-RL testbed.

**Rubric.** Our rubric contains the following 8 error types, spanning all of the events and consequences listed in Table 1.

1. When the ball hits the goal, it incorrectly bounces off.
2. When the ball hits the goal, the opponent score is incorrectly incremented.
3. When the ball hits the goal, no new ball is launched.
4. When the ball hits the floor, the opponent score is not incorrectly incremented.
5. When the ball hits the wall, the opponent score is incorrectly incremented.
6. When the left or right action is taken, the paddle moves in the wrong direction.
7. When the ball hits the paddle, the player score is incorrectly incremented.
8. When the program starts, no ball is launched.

Some errors in the program make it impossible to uncover other errors. For example, if no ball is launched at the start of the program, it is impossible to check whether error 7. is present in the program, because there is no ball to hit on the paddle. In these situations, we label all of the impossible to check errors as not present in the program.

Future work could investigate iteratively providing feedback on a program to a student in order to provide feedback about all errors, including those that are initially impossible to uncover. Such a system could work as follows: First, the grading system provides feedback about all of the errors it can currently find in the program. Then, the student updates their program to fix all of the errors found by the program. Finally, the student re-submits their updated program to the grading system for another round of evaluation. This process continues until there are no more errors in the program.

**Statistics.** The dataset consists of 711,274 submissions from 453,211 students — some students created multiple submissions. Amongst the 711,274 submissions, there are 111,773 unique programs. The error labels $y$ assigned to each program were programmatically generated by Nie et al. [30].

Averaged across the 6 programs and 9 human graders, humans achieve a raw inter-grader agreement of 85.4%. Agreement is computed as the fraction of label dimensions that all 9 human graders agree on.

**Recommended settings.** Though our code can flexibly support many settings, we recommend the following standard configurations to enable better comparison on future work on this dataset. We recommend that each episode last for 100 timesteps and terminate if the player or opponent score exceeds 30. We also recommend allowing $K$ episodes for exploring each program for bugs, where $K$ is the number of considered bugs (i.e., the dimensionality of the label).

# B DREAMGRADER Details

Our implementation of DREAMGRADER builds off the DREAM code released by Liu et al. [25] at https://github.com/ezliu/dream. We use the PyTorch [34] DQN implementation released by Liu et al. [24] and experimental infrastructure from Liu et al. [23].

## B.1 Model Architecture

DREAMGRADER consists of $K$ recurrent exploration policies $\{\pi_k\}_{k=1}^K$ and a feedback classifier $g(y \mid \tau)$.

**Exploration policies.** Each exploration policy $\pi_k$ is parametrized as a double dueling deep Q-network [28, 50, 45], consisting of a recurrent Q-function $Q(\tau_{:t}, a_t)$ and a target network $Q_{\text{target}}(\tau_{:t}, a_t)$ with the same architecture as the Q-function, where $\tau_{:t} = (s_0, a_0, r_0, \ldots, s_t)$ denotes the trajectory so far up to timestep $t$. To parametrize these Q-functions, we first embed the

Table 4: Hyperparameters used for DREAMGRADER.

| Hyperparameter | Value |
|---|---|
| Discount Factor $\gamma$ | 0.99 |
| Learning Rate | 0.0001 |
| Replay buffer batch size | 32 |
| Replay buffer minimum size before updating | 500 episodes |
| Target parameters syncing frequency | 5000 updates |
| Update frequency | 4 steps |
| Grad norm clipping | 10 |

trajectory $\tau_{:t}$ as $e(\tau_{:t}) \in \mathbb{R}^{64}$. Then, we apply two linear layers with output size 1 and $|\mathcal{A}|$ respectively. These represent the state-value function $V(\tau_{:t})$ and advantage $A(\tau_{:t}, a_t)$ respectively. Finally, following the dueling architecture [50], we compute the Q-value as:

$$Q(\tau_{:t}, a_t) = V(\tau_{:t}) + A(\tau_{:t}, a_t) - \frac{1}{|\mathcal{A}|} \sum_{a \in \mathcal{A}} A(\tau_{:t}, a). \tag{7}$$

To compute the trajectory embedding $e(\tau_{:t})$, we embed each tuple $(s_{t'}, a_{t'}, r_{t'}, s_{t'+1})$ for $t' = 0, \ldots, t - 1$ and then pass an LSTM [18] over the embeddings of the tuples. The embedding of $(s_{t'}, a_{t'}, r_{t'}, s_{t'+1})$ is computed by embedding each component and applying a final linear layer with output dimension 64 to the concatenation of the embedded components. We embed $s_{t'}$ and $s_{t'+1}$ with the same network, using the architecture from Nie et al. [30], consisting of two linear layers with output dimensions 128 and 64, respectively, with an intermediate ReLU activation. We embed the action $a_t$ with an embedding matrix with output dimension 16. We embed the scalar reward $r_t$ with a single linear layer of output dimension 32.

Each exploration policy $\pi_k$ is trained to maximize the expected discounted exploration rewards via standard DQN updates:

$$\mathcal{J}_{\exp}(\pi) = \mathbb{E}_{\mu \sim p(\mu), \tau \sim \pi(\mu)} \left[ \sum_{t=0}^{T} r_t^{\exp} \right], \tag{8}$$

$$\text{where } r_t^{\exp} = \log g(y_k \mid \tau_{:t+1}) - \log g(y_k \mid \tau_{:t}). \tag{9}$$

**Feedback classifier.** The feedback classifier $g(y \mid \tau)$ outputs a distribution over predicted labels $y \in \{0, 1\}^K$. We parametrize each dimension of the feedback classifier $g(y_k \mid \tau)$ with a separate neural network for simplicity. To parametrize $g(y_k \mid \tau)$, we embed the trajectory $\tau$ as $e(\tau)$ using a network similar to the one for the exploration policy. Then, we apply three linear layers with output dimensions 128, 128, and 2 respectively to $e(\tau)$ with intermediate ReLU activations. Finally, we apply a softmax layer to the output of the linear layers, which forms the distribution over $y_k \in \{0, 1\}$.

To embed the trajectory $\tau$ as $e(\tau)$, we embed each $(s_t, a_t, r_t, s_{t+1})$-tuple for each timestep $t$ in the trajectory $\tau$. Then, we pass an LSTM with output dimension 128 over the embeddings of the tuples, and take the last hidden state of the LSTM as $e(\tau)$. To embed each $(s_t, a_t, r_t, s_{t+1})$-tuple, we embed each component separately, and then apply two final linear layers with output dimensions 128 and 64 respectively and an intermediate ReLU activation. We use the same networks architectures used in the exploration policy trajectory embeddings to embed the state, action, and reward components.

Each dimension of the feedback classifier $g(y_k \mid \tau)$ is trained to maximize:

$$\mathcal{J}_{\text{feedback}}(g) = \mathbb{E}_{\mu \sim p(\mu), \tau \sim \pi(\mu)} \left[ \log g(y_k \mid \tau) \right]. \tag{10}$$

## B.2 Hyperparameters

We use the hyperparameters listed in Table 4 for all of our experiments. We chose values based on those used in Liu et al. [25] used for DREAM, and did not tune these values. We optimize the objectives written above using the Adam optimizer [22]. During training, we anneal the $\epsilon$ for $\epsilon$-greedy exploration from 1 to 0.01 over 250,000 steps. We use $\epsilon = 0$ during evaluations.

**Delayed updating.** At the beginning of training, the feedback classifier is randomly initialized and provides poor noisy reward signal for learning exploration. However, as the exploration policy occasionally visits states indicative of bugs, the feedback classifier begins to learn and improve its reward signal for the exploration policy. To ensure that the exploration policy only learns from good reward signal, we wait until the replay buffer contains at least 500 episodes before beginning to take gradient steps on the exploration policy, which is common in DQN-based algorithms. This allows time for the feedback classifier to learn before the exploration policy begins updating on the reward computed from the feedback classifier.

## C    Experiment Details

### C.1    Human Grading

**Grading details.** We obtained 9 volunteers to measure the grading accuracy of humans by soliciting volunteers from a university research group, consisting of computer science undergraduate, master's and PhD students involved in machine learning research. Each volunteer first received training about the Bounce programming assignment by reading a document describing the behavior of a correct Bounce program and all potential errors that may occur in incorrect implementations. Additionally, each volunteer was allowed to play a correct implementation for as long as they desired. After receiving training, each volunteer was presented with the same set of 6 randomly sampled Bounce programs, but in a randomized order. For each program, each volunteer was allowed to play the program for as long as they needed and was asked to list the errors they found in the program on a checklist including all possible errors. The reported human grading accuracy in Table 2 is the mean accuracy of the 9 volunteers on these 6 programs.

**Compensation.** Volunteers took 30–45 minutes total to familiarize themselves with the Bounce programming assignment and to grade the 6 programs. They were each compensated with a $10 gift card, amounting to a compensation of roughly $15 per hour.

**Institutional Review Board.** The grading process did not expose grading volunteers to any risks beyond that of normal life. It was reviewed by the Institutional Review Board (IRB) and it was determined that it did not constitute human subjects research and did not require IRB approval. Below is the final determination of the IRB.

> After further review, the IRB has determined that your research does not involve human subjects as defined in 45 CFR 46.102(f) and therefore does not require review by the IRB.

## D    Additional Results

### D.1    Results on Additional Error Types

Our experiments primarily focus on 8 error types that span all event and consequence types in Table 1 for simplicity. However, below, we include results of DREAMGRADER on all error types for completeness. There are 28 total error types, corresponding to the 6 event types times the 5 consequence types, minus 2 of the pairs — "when the paddle moves, the ball bounces / does not bounce" and "when the program starts, the ball bounces / does not bounce" — which are not found in student programs.

We train DREAMGRADER on the same $N = 3{,}556$ programs as the other experiments, but train and evaluate on all 28 error types. We find that DREAMGRADER's performance remains roughly the same as when it is trained and evaluated on only 8 error types. Specifically, DREAMGRADER achieves an average accuracy of 94% on all 28 error types, whereas it achieves an average accuracy of 94.3% on only 8 error types, and human accuracy is 95.8%. We did not find these results to be surprising, as the original 8 error types were already relatively representative, so we expected performance of the other error types to be similar.

## D.2 Results on Breakout

In this section, we include results on the Breakout assignment, an assignment widely taught in university and highschool classrooms. In this assignment, students are asked to program a Python game, which contains a ball and paddle, as well as several rows of bricks. The paddle should be able to move left or right, and the ball should bounce off the paddle, walls, and bricks. If the ball hits a brick, the brick should disappear. See Figure 4 for an example program.

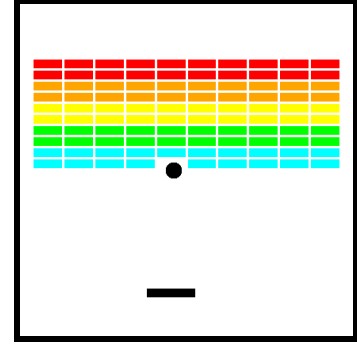

Figure 4: Breakout. One of the bricks was hit by the ball, creating a gap.

We did not have access to real student programs. However, we have the true rubric used to grade this assignment, which we have used to create 64 programs that include common student errors. We train DREAMGRADER on 32 of these programs and hold out the remaining 32 for testing. These errors include:

- *Paddle skewer.* The ball always reverses y-directions when it hits the paddle. This makes it possible to "skewer" the ball on the paddle, if the paddle hits the ball from the side as it is falling. When the ball makes contact with the paddle, it reverses directions, but may not move far enough in a single timestep to escape the paddle. Then, on the next timestep, it reverses directions again. This continues and the ball becomes stuck "skewered" on the paddle. In a correct implementation, the ball should only reverse y-directions when it hits the paddle if the ball is falling (and not if it is rising).
- *Does not delete brick.* The ball bounces off bricks, but the bricks do not disappear when hit.
- *Does not bounce off brick.* The ball incorrectly does not bounce off the bricks.
- *Wrong number of brick rows.* The assignment specifies that students should create 10 rows of bricks (to test their ability to program loops). It is incorrect if there are any number of rows of bricks not equal to 10.
- *Reversed paddle movement.* Pressing the left arrow key results in the paddle moving in any direction other than left, and similarly for the right arrow key.
- *Bounce off floor.* The ball incorrectly bounces off the floor, rather than falling out of screen.

We test DREAMGRADER on the error type that is most challenging for humans to grade, *paddle skewer*. This is challenging to grade because it requires careful timing to hit the ball on the side. We find that DREAMGRADER achieves an accuracy of 93.8% on the test programs, correctly grading 30 of the 32. DREAMGRADER learns to deliberately skewer the ball with the paddle, which can be seen at https://ezliu.github.io/dreamgrader. We also find that DREAMGRADER's performance is even higher on the less complex error types, and achieves 96.9% accuracy on the test programs for the *does not delete brick* error type.

**Experimental details.** Each Breakout episode consists of 300 timesteps and terminates early if all of the bricks are destroyed, or if the ball hits the floor. The reward function across all programs is 0 for all states and actions. The state consists of the $(x, y)$-coordinates of the ball, paddle, and bricks. We use the same DREAMGRADER architecture and hyperparameters that were used for the Bounce experiments.

## D.3 Bounce DREAMGRADER Exploration Behavior Visualizations

We visualize the exploration behaviors learned by DREAMGRADER's exploration policies in Figure 5. Qualitatively analyzing these behaviors shows that DREAMGRADER exploration behaviors that probe each possible event type listed in Table 1. Specifically, DREAMGRADER learns to hit the ball into the goal; hit the ball into the wall; hit the ball with the paddle; deliberately miss the ball; and move the paddle in various directions. Crucially, we find that behaviors are fairly robust to different programs. For example, we find that the exploration policies for bugs related to hitting the ball into the goal still successfully hit the ball into the goal most of the time, even when there are multiple balls, or when the actions are reversed, so that the left action moves the paddle right, and vice-versa. This

| Miss ball | Hit paddle / wall | Score goal | Test movement |
|:---:|:---:|:---:|:---:|
| 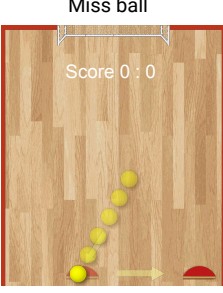 | 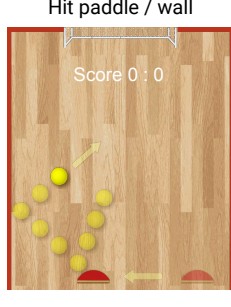 | 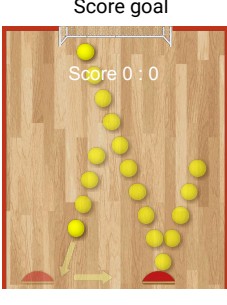 | 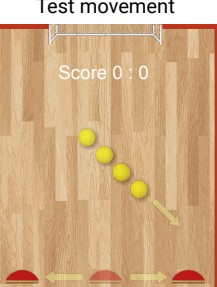 |

Figure 5: The exploration behaviors learned by DREAMGRADER. These exploration behaviors probe all of the possible events associated with errors, enabling DREAMGRADER to uncover all errors. Crucially, these exploration behaviors are robust to different programs. For example, the illustrated "score goal" exploration behavior succeeds in hitting a ball into the goal, even when there are multiple balls.

indicates that DREAMGRADER can handle the key challenge of our problem setting: learning diverse and adaptable exploration behaviors. Videos of DREAMGRADER's exploration behavior can also be found at https://ezliu.github.io/dreamgrader.

# E   Detailed Relation to Prior Work

## E.1   Connection with DREAM

In this section, we discuss the full connection between DREAMGRADER and DREAM. DREAM is a generic few-shot meta-RL algorithm that learns two components, an exploration policy, and an exploitation policy. Assuming that each of the meta-training tasks are distinguishable from each other with a unique problem ID (e.g., a unique one-hot), the DREAM exploitation policy learns to solve each task conditioned on an information-bottlenecked representation of the problem ID $z$, which aims to identify only the task-relevant information. Then, the exploration policy is learned to maximize the mutual information between exploration trajectories from rolling out the exploration policy, and bottlenecked representation $z$, which contains only the task-relevant information.

Primarily, DREAMGRADER leverages the exploration aspect of DREAMGRADER. Specifically, DREAMGRADER learns an exploration policy to maximize the mutual information between exploration trajectories and the label $y$, by using the same per-timestep dense reward decomposition that DREAM uses to maximize its mutual information objective. However, DREAMGRADER's feedback classifier can also be seen as a special-case of the DREAM exploitation policy: We can interpret exploitation episodes as single timestep episodes, where the feedback classifier takes a single action, i.e., predicting the label, and the reward is given as the number of label dimensions that are correctly predicted, and hence serves the same purpose as the exploitation policy. The difference is that whereas DREAM attempts to learn a representation $z$ that contains only the information needed from exploration by imposing an information bottleneck, $y$ already contains exactly that information, so no such bottleneck needs to be applied.

## E.2   Differences with Original Play-to-Grade Formulation

In this section, we discuss the primary high-level differences between DREAMGRADER and Nie et al. [30]. While both approaches aim to learn policies that discover errors in student programs by interacting with them, these approaches conceptually differ in how they frame this problem. Nie et al. [30] formulate the feedback challenge as computing the distance between two MDPs, a student program, and a correct reference program, and finding MDPs where this distance is over a threshold, indicating a buggy program. This leads to an approach, which involves learning a distance function between MDPs, based on how the dynamics and reward functions differ between the two MDPs, and an exploration policy to explore states where the dynamics and rewards significantly differ according to the distance function. However, Nie et al. [30] learn this distance function by learning dynamics and rewards models, which can be challenging, and result in poor reward signal for learning the exploration policy.

In contrast, this work connects the feedback challenge with the meta-exploration and meta-RL problem statements, which opens to door for applying meta-RL techniques. Specifically, DREAMGRADER leverages techniques from the DREAM meta-RL algorithm, which creates a dense and helpful reward signal for learning an exploration policy. Empirically, this reward signal results in much more effective exploration policies.

Overall, DREAMGRADER and Nie et al. [30] significantly conceptually differ in how they approach the feedback problem: Nie et al. [30] cast the problem as computing distances between MDPs, whereas DREAM casts the problem as meta-exploration, which results in superior performance.