# OpenReview forum: "Giving Feedback on Interactive Student Programs with Meta-Exploration"
_NeurIPS.cc/2022/Conference — NeurIPS 2022 Accept_

### Official Review · Reviewer_CtwK · 2022-07-09

**Rating:** 8
**Confidence:** 4
**Soundness:** 3 good
**Presentation:** 2 fair
**Contribution:** 4 excellent

**Summary:**

The paper presents a novel approach called DreamGrader to give feedback on
interactive student programs. In contrast to previous approaches which only gave
binary feedback (correct/incorrect), they assign more fine-grained feedback from
an error rubric of 8 possible errors. They tackle giving feedback as a
meta-exploration problem where each program defines an MDP, and the task is to
explore these efficiently to predict the errors accurately. For this both an
exploration policy and a feedback classifier is learned.

The exploration policy is learned by maximizing the mutual information between
the exploration trajectories and the feedback labels, similarly to how Dream
maximizes the mutual information between the trajectories and the task-relevant
information. Following Dream further, this objective is decomposed according to
the timesteps to yield a shaped reward function which captures how much new
information a transition provides to the feedback classifier on the errors.


**Questions:**

Table 1 shows $6 \times 5 = 30$ possible error cases, of which 8 are included in
the rubric. I find that some possibly important cases are missing like "When the
ball hits the wall, it incorrectly doesn't bounce". What was the reason for
constraining the error rubric to 8 errors? Could the method scale to 30 error
cases?

In Figure 3, I find the human baselines surprisingly low in some cases. For
example, I don't see how a human grader could miss the "No Ball at Start" error,
as it should be obvious after launching the game. Could you explain?

This is more of a curiosity question: The method works on programs that are
MDPs. What can we do when this assumption does not hold?


**Limitations:**

I cannot think of any limitations which are not addressed.


**Strengths And Weaknesses:**

## Strengths

As far as I know, the method is a first of its kind, able to give fine-grained
feedback on interactive programs that correspond to MDPs automatically. It could
have significant impact in computer science education research and applications.

## Weaknesses

I think the main weakness of the paper is that its connection to the Dream
meta-RL algorithm is not clear enough. In my understanding, the main
contribution of Dream is Decoupling the exploRation and ExploitAtion in Meta-RL
(if we read the capitalized letters together we get Dream).
However, the present paper does not make use of this concept, because we already
have the labels for the programs (those correspond to the task-relevant
information in Dream), so it needs to utilize only the exploration part of Dream
along with training the classifier.

Moreover, Section 4.3, which should connect the presented method to Dream does
not really describe Dream but meta-RL in general, which is confusing, as
exploration and exploitation works somewhat differently in Dream.

I think the connection between Dream and the presented method should be much
clearer, and the authors could maybe consider it naming something different
instead of DreamGrader.

Even though the play to grade paper [25] also included just one non-toy task, it
would be good if the results were demonstrated on more than one programming
task.

I find it somewhat of an exaggeration that changing the speed of the ball and
paddle are examples of creativity.

Small typo: line 269 - time time

---

> ### Author Response · Authors · 2022-08-02
> **Author response**
>
> We thank Reviewer CtwK for the helpful and thoughtful review. We have strengthened the paper according to the reviewer's suggestions and summarize below.
>
> Reviewer CtwK’s main concern is about clarifying how this **work connects with DREAM and meta-RL.** Specifically, Reviewer CtwK correctly notes that we only leverage the exploration part of DREAM to convert our mutual information objective into a per timestep reward for learning the exploration policy. DreamGrader can be seen as a special-case of DREAM, where the task-relevant information z is already known during training and is equal to the label y: We can view the exploitation episode as a single timestep of predicting the label, with rewards proportional to the number of dimensions of the label that is correctly predicted. Then, given the label y, the optimal exploitation policy is just the identity function (outputting y), and the only challenge is learning the exploration policy. In other words, DreamGrader leverages DREAM to learn the exploration policy and DreamGrader’s feedback classifier is exactly the DREAM exploitation policy, but without the need to extract the task-relevant information z.
>
> Reviewer CtwK also asks **why Section 4.3 connects our setting with meta-RL in general, rather than DREAM specifically.** Section 4.3 shows how our problem of learning a policy to discover bugs in a program relates to the general meta-exploration problem in meta-RL, which provides motivation for leveraging ideas from DREAM: DREAM is a meta-RL algorithm designed to tackle meta-exploration, the key challenge in our setting. This section connects the setting with meta-RL in general, rather than DREAM specifically in order to offer the setting as a testbed for future meta-RL research. While our specific DREAM-based approach performs well, the setting is ripe for exploring other meta-RL algorithms and ideas. However, we are happy to revise and clarify any parts that are confusing.
>
> **Why does the paper focus on 8 error types? “Could [DreamGrader] scale to 30 error cases?”** We focused on a subset of 8 error types in the paper to reduce the amount of compute. However, these error types spanned all possible action and consequence types to ensure that they covered all representative bugs. During the rebuttal period, we have run a single seed of DreamGrader on all 30 error types and will include these results in the paper when we have completed more seeds. We find that DreamGrader achieves roughly the same average accuracy across all 30 error types, as it does on just the 8.
>
> |                              | Average Accuracy |
> |------------------------------|:----------------:|
> | Human (8 error types)        |       95.8%      |
> | DreamGrader (8 error types)  |       94.3%      |
> | DreamGrader (30 error types) |       94.0%      |
>
> **“It would be good if the results were demonstrated on more than one programming task.”** We have recently obtained access to a second assignment called Breakout, which is widely taught in university and highschool classrooms. In this assignment, students are asked to program a Python game, which also contains a ball and paddle, as well as several rows of “bricks,” where the object of the game is to break all the bricks by hitting them with a ball. We are unable to build the necessary infrastructure for full experiments on this assignment during the rebuttal period. However, we hope to offer preliminary results on this assignment near the end of the discussion period.
>
> **“I find it somewhat of an exaggeration that changing the speed of the ball and paddle are examples of creativity.”** We agree with this concern and have revised Section 5.3 to frame this as generalizing to unseen behaviors during training, rather than calling it creativity.
>
> **Human results in Figure 3.** Figure 3 in the submission uses the same human performance line in all of the plots, equal to the overall average human accuracy (95.8%). We’ve updated the draft so that the line in each plot reflects the human accuracy on that specific error type and adjusted the text accordingly. In the updated plots, humans achieve 100% accuracy on the obvious error “No Ball at Start.”
>
> **Can DreamGrader apply to programs that are not MDPs?** Since the reward signal for the exploration policy is non-Markov, the exploration policy is already parametrized to condition on the entire history of states, actions, and rewards, rather than the current state. In principle, the exploration policy is therefore suitable for learning over partially-observable Markov decision processes (POMDPs), though we acknowledge that partial observability can make learning the exploration policy practically more challenging.
>
> Please let us know if there are any remaining questions or concerns!

---

> > ### Comment · Reviewer_CtwK · 2022-08-07
> > **Thank you for your thoughtful answers**
> >
> > I'm particularly happy about the performance of the method across all the 30 error types, it's very promising.
> >
> > I agree with most of your answers, and I see how you could look at the feedback classifier as the exploitation policy of Dream, even though I find it's still a bit of a stretch. It would be good to explain this also in the paper.
> >
> > I increased my score to Strong Accept.

---

> > > ### Author Response · Authors · 2022-08-08
> > > **Thank you for the response!**
> > >
> > > We thank Reviewer CtwK for their response and time. We have added the discussion about how the feedback classifier can be seen as a special case of the DREAM exploitation policy to the paper under Appendix E.1. We are also willing to change the name of the approach to something that does not reference DREAM, if this connection is a major issue.
> > >
> > > Additionally, Reviewer CtwK previously asked if we could **experiment on an additional assignment.** We have completed the preliminary results on the Breakout assignment that we mentioned in our previous response. In these results, we did not yet gain access to real student programs, but we have spoken to an instructor and have the true rubric used to grade this assignment, which we used to create 64 programs that include common student errors from the rubric, split into 32 programs for training and 32 for testing. Each of these programs is 300+ lines of Python code.
> > >
> > > We tested DreamGrader on the hardest error type for humans to grade: skewering the ball with the paddle. This error arises when the student always reverses the ball’s y-velocity whenever it hits the paddle. Then, if the paddle moves sideways into the ball as it is falling, the ball changes directions to move upwards but may not move far enough in a single timestep to escape the paddle. This causes the ball to reverse y-direction again (now down). This repeats with the ball stuck “skewered” on the paddle. This error is hard and frustrating for humans to grade, because it requires careful timing to hit the ball sideways with the paddle. **We find that DreamGrader achieves 93.8% accuracy on detecting this error.** We’ve included these results with more details in Appendix D.2 of the paper, and will include additional results as we run more experiments on the other error types and obtain real student programs. The supplementary submission also now includes a visualization of a DreamGrader exploration policy skewering the ball in a buggy program under skewer_paddle.gif.

---

> > > > ### Comment · Reviewer_CtwK · 2022-08-08
> > > > **Thank you for your response**
> > > >
> > > > The experiments with breakout are very interesting, especially as I have seen exactly this error comitted at class.
> > > >
> > > > I do not consider referencing Dream in "DreamGrader" a serious issue, especially after your response.

---

### Official Review · Reviewer_BUXC · 2022-07-09

**Rating:** 6
**Confidence:** 3
**Soundness:** 3 good
**Presentation:** 3 good
**Contribution:** 3 good

**Summary:**

This paper proposes a meta exploration reinforcement learning algorithm, dreamgrader, that tries to provide feedback to student-authored interactive programs. The authors conducted experiments on a large real dataset and showed that their approach is effective.

**Questions:**

1. If I understood correctly (I may not have), an important benefit of meta learning is generalization to new tasks. A task can be a new student program, which is what this paper studies in its experiments, for sure, but it can be a new error type? What if you leave out an error during training, train on only 7 errors, and see if your learned policy can adapt to the 8th? Like Figure 7 (middle) in the DREAM paper? A discussion or an additional experiment would help a lot here.

2. Following the previous question - the factorized version of the method learns a separate exploration policy and a separate feedback classification model for each error. This means 8 different sets of models. I don't know if this is any different than the original play to grade approach (1 binary label) since you can treat each error label independently. Which means fundamentally I don't see the improvement here over that paper: feedback simply means doing 8 separate binary classification tasks, so can we do the same with the previous approach just repeated 8 times?

3. Following the previous question - why are the errors a ordered list (line 121) but treated independently? Intuitively, I guess some errors could be related, which means there could be some shared layers in these models (e.g., for trajectory representations) before going their separate ways. Is there no sweet spot between one unified model for all errors and 8 separate models?

4. The baseline is basically a strawman (one that cannot leverage more than 8 training data points). It's still good to see the direct max version, though, since that illustrate the benefit of credit assignment (which I don't know is the new thing in this paper compared to DREAM).

5. More experimental details can help: you claim super human performance on some errors, which is understandable, but what is human interreater agreement? How are the ground truth on the 99.5% of the programs not seen during training obtained? Each grader only graded 6 problems and those are averaged in Table 2? Are the results reliable with such a small evaluation set? Any statistical significance?

**Limitations:**

The authors did discuss potential limitations in Section 6.

**Strengths And Weaknesses:**

Strength:
- The overall idea is good: using meta RL to learn how to balance exploration and exploitation for different errors when grading programs
- Experiments are conducted on a large, real dataset, not a synthetic one

Weaknesses:
- Some details of the experiments are unclear. But hopefully this can be resolved through discussions, I'm happy to revisit the rating again depending on what the authors and other reviewers say
- The connection to meta reinforcement learning needs more discussion (essentially only 2 sentences lines 273-275). More discussions would help

---

> ### Author Response · Authors · 2022-08-02
> **Author response (part 1 of 2)**
>
> We thank Reviewer BUXC for their time and thoughtful comments, and we have accordingly updated and strengthened the paper.
>
> Reviewer BUXC requests additional discussion on the **connection to meta-RL.** The standard few-shot meta-RL setting consists of 1) sampling a new task; 2) exploring the task for a few episodes (i.e., the few shots) to gather information; 3) leveraging the explored information to achieve high returns on new episodes. Our setting exhibits the same structure: at test time, 1) a new student program is given to the agent; then the agent is allowed to 2) explore the program for a few episodes to find bugs; and it must use this information to 3) predict the label (i.e., output what bugs are in the program). Effectively, the final evaluation episodes that occur during 3) are 1-step problems where the action space is predicting the label, and the reward is the accuracy of the predicted label. Hence, the setting of providing feedback can be seen as a form of meta-RL or meta-bandits where there is a supervised label available during training. We have added this discussion to Section 4.3.
>
> **Questions**
>
> 1. **Can you generalize to new error types at test time?** Even human graders are unable to generalize to new error types without being shown examples of the error or being told what the new error means. Hence, generalizing to new error types without additional training programs is generally not possible. However, this is not problematic as the main goal in this setting is to generalize to new student programs, rather than new error types, since the rubric is typically relatively static. Additionally, if new errors are added, another policy can simply be trained on the new error types, although this requires training data.
>
> 2. **How does DreamGrader differ from Nie et al., ‘21?** DreamGrader and Nie et al., ‘21 both seek to learn an exploration policy that discovers bugs in the student program, but critically differ in how this policy is learned. This difference accounts for DreamGrader’s superior performance. Both approaches seek to create a reward signal for the exploration policy that is high when an error is uncovered. Nie et al., ‘21 attempts this by assuming access to a correct reference program and learning a reward $\hat{R}(s, a)$ and dynamics model $\hat{T}(s, a)$ of this reference program. Then, the exploration policy is given high reward for visiting (state s, action a, reward r, next state s’)-tuples, where either the reward r or next state s’ highly differ from those predicted under the learned models. However, learning accurate reward and dynamics models is challenging, which can result in noisy or poor reward signal for the exploration policy. In contrast, DreamGrader formalizes the same intuition by learning to maximize the mutual information between the label and the trajectories produced by the exploration policy, which is practically achieved with the decomposition from DREAM. This approach avoids challenges of learning accurate models, and provides significantly better reward signal for the exploration policy.
> **To provide fine-grained feedback, can’t we just use Nie et al., ‘21 independently on each of the error types?** We agree that the most natural way to extend Nie et al., ‘21 to the fine-grained feedback setting is by applying it independently on each of the 8 error types. In fact, that is effectively what the Nie et al., ‘21 extension that we compare with does.

---

> > ### Author Response · Authors · 2022-08-02
> > **Author response (part 2 of 2)**
> >
> > 3. **Could you share representations between policies for different errors?** Yes, many of the policies and feedback classifiers likely could significantly share parameters. For example, the policy that tests if hitting the ball into the goal creates a new ball could likely also be used for testing if hitting the ball into the goal increments the player score. However, we do not assume knowledge of which errors are related to each other, and therefore choose not to share any parameters for simplicity. We acknowledge that more complex schemes could be used to share parameters without knowing which errors are related to each other, but we chose to use the simplest scheme that works. We’ve added this discussion in Section 4.2.
> >
> > 4. **“The [extension of Nie et al., ‘21] baseline is basically a strawman.”** We believe there is a misunderstanding here. The Nie et al., ‘21 extension that we use is the most natural extension of the original Nie et al., ‘21 approach to the fine-grained feedback setting, where we apply the approach independently for each error type, which was exactly the reviewer’s suggestion in question 2. It is true that the Nie et al., ‘21 approach is unable to incorporate many training programs, but that is a drawback of the original Nie et al., 21 approach, which is the state-of-the-art approach for the coarse-grained feedback setting. If the reviewer had an alternate suggestion of an appropriate baseline that we overlooked, we would welcome the suggestion.
> >
> > 5. **Additional experimental details**
> > - **“What is the human inter-grade agreement?”** The raw inter-rater agreement is 85.4%, defined as the fraction of errors where all 9 human graders agreed, averaged across all error types and the 6 graded programs.
> > - **How are the ground-truth labels for the programs obtained?** We use the labels released by Nie et al., ‘21, which are programmatically computed with a hard-coded script that checks for conditions in the student programs. We note that Bounce was chosen as a testbed because it was possible to automatically obtain these ground-truth labels, but DreamGrader applies to and is most valuable on assignments where automatically obtaining these labels is challenging.
> > - We confirm that each human grader only graded 6 programs, as human grading is too slow to evaluate over large sets. However, the results in Table 2 for the automated approaches are averaged over 1,000 sampled test programs, which holds statistical significance.
> > These details are included in Appendix A.
> >
> > Reviewer BUXC states that their main concern is clarifying details about the experiments and that they are willing to reconsider their score once these are clarified. Please let us know if there are any remaining details about the experiments that you would like clarified!

---

> > > ### Comment · Reviewer_BUXC · 2022-08-08
> > > **I read the rebuttal**
> > >
> > > I think the rebuttal helped my understanding, although my opinion largely remains unchanged - to me, 30 error types and 8 error types are not any different. Moreover, the real challenge is in situations where ground-truth labels cannot be obtained and extensive human grading effort is necessary, which is the premise of the paper, however the experiments are conducted in a situation where labels can be automatically obtained. Nevertheless, this is a solid paper although the improvement over Nie et al., 2021 is incremental from my point of view. If other reviewers think highly on this paper, I'm happy to concur.

---

> > > > ### Author Response · Authors · 2022-08-08
> > > > **Thank you for the response!**
> > > >
> > > > We thank Reviewer BUXC for their time and response. We respond to their main remaining concerns below.
> > > >
> > > > **Additional results on an assignment where ground-truth labels cannot be obtained automatically.** Following the other reviewers’ requests, we have completed preliminary results on a second assignment, called Breakout. On this assignment, **it is not possible to automatically obtain the ground-truth labels**. We have not yet gained access to real student programs, but we have spoken to an instructor and have the true rubric used to grade this assignment, which we used to create 64 programs that include common student errors from the rubric, split into 32 programs for training and 32 for testing. Each of these programs is 300+ lines of Python code.
> > > >
> > > > We tested DreamGrader on the hardest error type for humans to grade: skewering the ball with the paddle. This error arises when the student always reverses the ball’s y-velocity whenever it hits the paddle. Then, if the paddle moves sideways into the ball as it is falling, the ball changes directions to move upwards but may not move far enough in a single timestep to escape the paddle. This causes the ball to reverse y-direction again (now down). This repeats with the ball stuck “skewered” on the paddle. This error is hard and frustrating for humans to grade, because it requires careful timing to hit the ball sideways with the paddle. **We find that DreamGrader achieves 93.8% accuracy on detecting this error.** We’ve included these results with more details in Appendix D.2 of the paper, and will include additional results as we run more experiments on the other error types and obtain real student programs. The supplementary submission also now includes a visualization of a DreamGrader exploration policy skewering the ball in a buggy program under skewer_paddle.gif.
> > > >
> > > > Reviewer BUXC’s other main concern is about **whether the work is incremental over Nie et al., ‘21.** We emphasize that a main contribution of this work is connecting the feedback setting with meta-exploration. This opens the door for meta-RL based methods, like DreamGrader, which significantly outperform straightforward extension of Nie et al., ‘21. Framing the problem in terms of meta-exploration is significantly different from the problem formulation in Nie et al., ‘21, which frames the problem as computing the distance between two MDPs. We believe this new connection with meta-RL is not incremental, though we are happy to further discuss and clarify, and we have added this discussion to Appendix E.2.
> > > >
> > > > Thank you for being active in the discussion period!

---

### Official Review · Reviewer_DZH4 · 2022-07-10

**Rating:** 8
**Confidence:** 4
**Soundness:** 3 good
**Presentation:** 4 excellent
**Contribution:** 3 good

**Summary:**

In this paper, the authors present DreamGrader, an automated fine-grained grading system for identifying known errors in interactive programs. DreamGrader consists of an program-interaction agent and an error-classifier. The agent interacts with the program and the classifier identifies the presence of different errors based on the agent interaction. The agent and classifier are learnt from a dataset of programs with error annotation. Towards this end, the authors present and leverage a compelling connection between error discovery and meta-reinforcement learning, leading to a unified object for simultaneously learning both the agent and the classifier. The proposed system outperforms strong baselines and prior work by a significant margin.

**Questions:**

## Questions

**Q1.** **Meta-RL benchmark**: Can the authors provide more details regarding the benchmark?

**Q2.** Since the classifier is used to generate the rewards for the exploration agent, how is cold-start problem (i.e. an under trained classifier would provide poor reward signals as well) resolved?

 **Q3.** Regarding Figure 3. In plot 1 the performance decreases between time-steps 4000-5000. That is quite curious. Is there any explanation for this behavior?

**Q4.**  Regarding Figure 3: Why are all the human performance line at the same level? Since they return the list of found errors (cf. appendix C.1) shouldn't the plot be different for at least some error-types?

**Q5.** Section 5.3: the result of having equally poor performance on training set speeds as on test set speeds indicates perhaps more data could have resolved the issue. Have the authors tried to have more data for this experiment? Does it change the results?

**Q6.** Do the authors have any explanation for why the extension of [1] performs quite poorly?

## Suggestion:

Making Figure 3 color-blind friendly would be nice! Further, it would be valuable to have the y-axis annotated on all the graphs.

## References

[1] Play to Grade: Testing Coding Games as Classifying Markov Decision Process, Allen Nie et al., NeurIPS 2021

**Limitations:**

Yes, the authors have adequately addressed the limitations and potential negative societal impact of their work.

**Strengths And Weaknesses:**

# Strengths:

1. **Originality**: The key insight of the authors, i.e., to pose the known-error discovery problem as an exploration-in-meta-RL problem, is original and novel. This work opens this research direction for further investigation.

2. **Presentation**: The paper is very well written. I really appreciate the authors for providing a well grounded motivation for the problem at hand (providing fine-grained feedback on interactive program assignments).

3. **Impact**: The proposed system is fairly feasible to deploy, specially in MOOCs, and can lead to impactful improvements in student grading in massive open online courses (MOOCs).

4. **Experiments**: The proposed method is compared against a power extension of prior work ([1]) and surpasses it by a large margin. Further, the results on real-world programs rather than simulated ones, which is always a plus point.

# Minor Weaknesses:

1. The proposed method is verified for only a single assignment. While this is due to the lack of annotated datasets, it does raise questions about the generality of the results.

2. **Meta-RL benchmark**: The paper espouses the use of its student-program dataset as a meta-exploration benchmark, it doesn't show thorough details (for example, proper statistics of the dataset or performance of baselines).  While the authors can be commended for releasing the code, simply releasing the dataset is not sufficient to call it a benchmark, *and* to count it as a contribution.

3. The approach is tested on very simple settings - 8 error-types and 100 time-steps. Interactive programming assignments typically can consist of an order of magnitude more types of error and number of time-steps. The method would become inapplicable if data requirements also grow proportionally.

## References

[1] Play to Grade: Testing Coding Games as Classifying Markov Decision Process, Allen Nie et al., NeurIPS 2021

---

> ### Author Response · Authors · 2022-08-02
> **Author Response (part 1 of 2)**
>
> We thank Reviewer DZH4 for their thoughtful review and suggestions. We have strengthened the paper by incorporating their feedback and summarize below.
>
> **Meta-RL Benchmark.** Reviewer DZH4 raises concerns about claiming a benchmark contribution. We agree with this concern and have updated the paper to clarify that the key contribution is connecting the problem of providing feedback with the meta-RL problem. We release the task as a testbed to support further meta-RL and education research, but do not claim to propose a new meta-RL benchmark. Additionally, DZH4 asks for more details about the task, which we include below and in Appendix A. If there are any other details that DZH4 would find useful, we are happy to include them in the discussion period.
> - The dataset consists of 711,274 student submissions from 453,211 students with 111,773 unique programs.
> - In our suggested settings, each episode lasts for 100 timesteps and terminates if either the player score or opponent score exceeds 30. We also suggest a standard meta-RL evaluation of allowing $K$ episodes for exploring each new program, where $K$ is the number of considered bugs (i.e., the dimensionality of the label).
> - Averaged over the 6 graded programs with 9 human graders, the average raw inter-grader agreement is 85.4%. Agreement is computed as the fraction of label dimensions that all 9 human graders agree on.
>
> **Generality.** We agree with Reviewer DZH4 that it is always preferable to have experiments on multiple settings. We have recently obtained access to a second assignment called Breakout, which is widely taught in university and highschool classrooms. In this assignment, students are asked to program a Python game, which also contains a ball and paddle, as well as several rows of bricks, where the object of the game is to break all the bricks by hitting them with a ball. We are unable to build the necessary infrastructure for full experiments on this assignment during the rebuttal period. However, we hope to offer preliminary results on this assignment near the end of the discussion period to test generality.
>
> **Scalability.** Reviewer DZH4 wonders how the data requirements scale to assignments with more error types and longer horizons. To test scalability to more error types, we’ve added experiments on Bounce with the full set of 30 error types. We find that using the same 3,556 training programs, DreamGrader achieves high accuracy across all 30 error types (see table below), so no additional data is required to scale to more error types in Bounce. We were only able to run a single seed during the rebuttal period, but we are currently running more, and we will update the paper with these results once all seeds complete.
>
> |                              | Average Accuracy |
> |------------------------------|:----------------:|
> | Human (8 error types)        |       95.8%      |
> | DreamGrader (8 error types)  |       94.3%      |
> | DreamGrader (30 error types) |       94.0%      |
>
> Our preliminary experiments on Breakout will test scalability to longer horizons as a Breakout episode is longer than that of Bounce, because it involves hitting up to 10+ bricks. We hope to report these results when they are available near the end of the discussion period.

---

> > ### Author Response · Authors · 2022-08-02
> > **Author response (part 2 of 2)**
> >
> > **Questions:**
> > - **“How is the cold-start problem resolved?”** Roughly, the feedback classifier learns first, and then later begins to output useful reward signal for the exploration policy. More specifically, even at the beginning of training, the exploration policy manages to occasionally visit states that either indicate or rule out bugs via random epsilon-greedy exploration, which enables the feedback classifier to learn. Then, as the feedback classifier learns, it outputs useful reward signals for learning the exploration policy. To prevent the exploration policy from overfitting to the initial poor reward signal from the feedback classifier, we do not begin updating the policy until the replay buffer contains a minimum number of episodes (set to be 500), after which point, the feedback classifier is more likely to provide useful reward signal. We have included this discussion in Appendix B.
> > - **“Why does performance drop at 4-5K steps in plot 1 of Figure 3?”** This appears to simply be standard instability common in deep RL algorithms. For example, [1] shows even more severe drops. This can be mitigated by terminating training when reward is high, or in this particular instance, the performance returns back to higher levels after further training (not shown in the plot).
> > - **“Why are all the human performance lines at the same level [in Figure 3]?”** We thank the reviewer for pointing this out. Figure 3 in the submission uses the same human performance line in all of the plots, equal to the overall average human accuracy (95.8%). We’ve updated the draft so that the line in each plot reflects the human accuracy on that specific error type and adjusted the text accordingly. We’ve also updated the plot to include y-axis ticks and colorblind-friendly colors, as suggested.
> > - **What happens when you increase the amount of training data on the experiments with varied ball and paddle speeds?** We ran additional experiments, where we doubled the amount of training data to ~7K programs on the varying ball and paddle speed setting. We find that performance does not appreciably change, suggesting that the lower performance is not simply from not enough data. Instead, qualitatively inspecting the errors suggests that the issue is that on the fastest ball speeds, sometimes it is not possible to move the paddle quickly enough to hit the ball, and hence some bugs cannot be discovered. We have added this to Section 5.3.
> > - **“Why [does the] extension of [Nie et al., ‘21] perform quite poorly?”** We note that the Nie et al., ‘21 approach is unable to leverage large amounts of training programs, and only trains with 10 curated programs. This appears to hinder its ability to explore effectively in new test programs exhibiting behavior not seen during training (e.g., programs with multiple balls). This is less of a problem in the coarse-feedback setting considered by Nie et al., ‘21, as determining that a test program exhibits behavior that does not match a reference program is easier than deliberating probing the test program to find exactly in what ways the test program differs.
> >
> > [1]: Averaged-DQN: Variance Reduction and Stabilization for Deep Reinforcement Learning. Oron Anschel, Nir Baram, Nahum Shimkin.

---

> > > ### Comment · Reviewer_DZH4 · 2022-08-06
> > > **Thank you for the comments!**
> > >
> > > Thank you for addressing the concerns I raised regarding Meta-RL benchmark, Generality and Scalability. The results on 30 errors (vs 8 in the earlier draft) is very promising. I also found the answers to my questions to be very reasonable.
> > >
> > > As reviewer BUXC and CtwK point out, the link to Dream meta-RL can be a little tenuous (i.e. the work doesn't fully exploit the different phases of Dream - in effect, the solution formulation is closer to meta-RL, rather than Dream meta-RL). The authors do provide a rebuttal to this argument, and I will be curious to hear the other reviewer's view on the author's reply.
> > >
> > > I am inclined to retain my score, as I believe the paper elegantly presents a novel solution for an interesting problem, and may have a strong impact by paving the way for further development of automated fine-grained feedback systems.

---

> > > > ### Author Response · Authors · 2022-08-08
> > > > **Thank you for the response!**
> > > >
> > > > We thank Reviewer DZH4 for their response and continued support of this work.
> > > >
> > > > **Preliminary results on a second assignment.** We’ve completed the preliminary results on the Breakout assignment that we mentioned in our previous response. In these results, we did not yet gain access to real student programs, but we have spoken to an instructor and have the true rubric used to grade this assignment, which we used to create 64 programs that include common student errors from the rubric, split into 32 programs for training and 32 for testing. Each of these programs is 300+ lines of Python code, and the horizon is 3x longer than that of Bounce, terminating after 300 steps.
> > > >
> > > > We tested DreamGrader on the hardest error type for humans to grade: skewering the ball with the paddle. This error arises when the student always reverses the ball’s y-velocity whenever it hits the paddle. Then, if the paddle moves sideways into the ball as it is falling, the ball changes directions to move upwards but may not move far enough in a single timestep to escape the paddle. This causes the ball to reverse y-direction again (now down). This repeats with the ball stuck “skewered” on the paddle. This error is hard and frustrating for humans to grade, because it requires careful timing to hit the ball sideways with the paddle. **We find that DreamGrader achieves 93.8% accuracy on detecting this error.** We’ve included these results with more details in Appendix D.2 of the paper, and will include additional results as we run more experiments on the other error types and obtain real student programs. The supplementary submission also now includes a visualization of a DreamGrader exploration policy skewering the ball in a buggy program under skewer_paddle.gif

---

> > > > > ### Comment · Reviewer_DZH4 · 2022-08-08
> > > > > **Exciting!**
> > > > >
> > > > > That sounds very exciting and promising! This error does seem hard to detect. I'd be curious to know about the performance of the extension of Nie et al. in this setting (though, I'd expect it to be abysmal). Kudos to the authors for pushing the results further!

---

### Author Response · Authors · 2022-08-02
**Author response summary**

We thank all reviewers for their time and helpful reviews. We’ve responded to each review separately, and we believe that we have addressed all reviewer concerns and questions. Please let us know if there are any other questions or concerns during the discussion period. Thank you!

We have strengthened the paper with the suggestions made by the reviewers and summarize the main changes below:
- We have run new experiments on all 30 error types. We have reported the results from the single seed that has finished running in the author responses, and will update the paper when all seeds are complete.
- We have included new experiments on increasing the amount of training data in the experiments that test generalization to new ball and paddle speeds.
- We have clarified the connection with meta-RL in Section 4.3.
- We have added more experimental details in the experiments sections and in the Appendix.

---

### Meta-Review · Area_Chair_J1c9 · 2022-08-25

**Recommendation:** Accept
**Confidence:** Certain

**Metareview:**

This work presents dreamgrader that aims to provide feedback to student-authored interactive programs. Reviewers all agreed that this paper presents a novel and original idea, solid experiments on real-world programs, as well as potential impact on MOOCs. There were some minor concerns and most got resolved during the discussion stage. Thus we recommend acceptance.


**Award:**

Yes

---

### Decision · Program_Chairs · 2022-09-14

Accept